# Diverse Diffusion: Enhancing Image Diversity in Text-to-Image Generation

## Abstract

Latent diffusion models excel at producing high-quality images from text. Yet, concerns appear about the lack of diversity in the generated imagery. To tackle this, we introduce Diverse Diffusion, a method for boosting image diversity beyond gender and ethnicity, spanning into richer realms: diversity for birds, butterflies, and, actually, quite arbitrary prompts.

Diverse Diffusion is a general unsupervised technique that can be applied to existing text-to-image models. Our approach focuses on finding vectors in the Stable Diffusion latent space that are distant from each other. We generate multiple vectors in the latent space until we find a set of vectors that meets the desired distance requirements and the required batch size.

To evaluate the effectiveness of our diversity methods, we conduct experiments examining various characteristics, including color diversity, LPIPS (Zhang et al., 2018) metric, and ethnicity/gender representation in images featuring humans. We also provide image quality assessment by human raters.

The results of our experiments emphasize the significance of diversity in generating realistic and varied images, offering valuable insights for improving text-to-image models. Through the enhancement of image diversity without decrease in quality, our approach contributes to the creation of more inclusive and representative AI-generated art.

## 1 Introduction

Latent diffusion models have gained significant attention for text-to-image generation, with DALL-E (Ramesh et al., 2021) and Stable Diffusion (Rombach et al., 2022) being some of the most prominent examples. While they have shown impressive results in generating high-quality images from textual descriptions, there have been concerns regarding a diversity loss.

Bias can arise from various sources, such as the training data or the algorithmic biases inherent in the model architecture. Struppek et al. (2022) shows that text-to-image models pick up cultural biases linked to various Unicode scripts. While increasing training data quality remains a significant concern and solutions such as (Smith et al., 2023) have been proposed to tackle dataset level bias, it is also important to ensure that diffusion-based methods generate diverse outputs that do not amplify the biases present in the training data.

The lack of diversity problem was addressed in (Ho, 2023), (Bianchi et al., 2023) and (Fraser et al., 2023). There, the authors notice that images generated by Stable Diffusion lack diverse cultural representation and are prone to gender stereotypes. Berns (2022) highlights the need for algorithmic adjustments in generative models to increase the diversity of their output for multi-solution tasks. It also proposes a framework that integrates automated machine learning with computational creativity to automate key tasks in artistic pipelines and increase the creative autonomy of computational agents. Further, Theron (2023) argues that Stable Diffusion v1-4 violates demographic parity in generating images of a doctor given a gender- and skin-tone-neutral prompt. The author observed that the model is biased towards generating images of perceived male figures with lighter skin, with a significant bias against figures with darker skin, as well as a notable

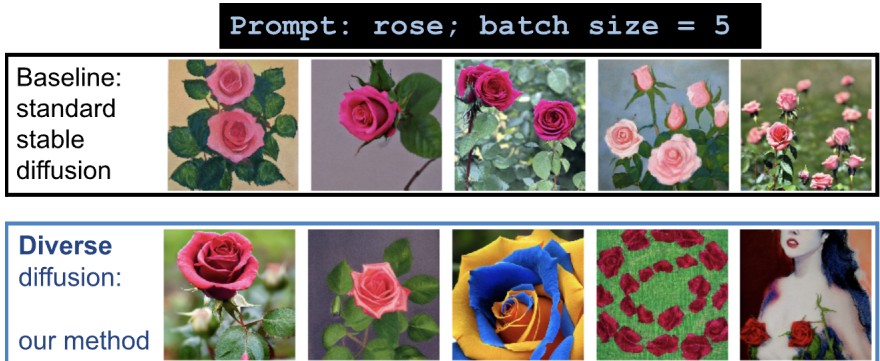

Figure 1: Images generated with standard Stable Diffusion and our diversity enhancement method for the prompt "rose", batch size = 5. Here, our method is shown to select images not only diverse in colors, but also in ideas, compared to vanilla Stable Diffusion.

bias against perceived female figures. According to Barr (2022), AI image generators often display gender and cultural biases. Stable Diffusion as other models has inherent biases from the training datasets. It was found, for instance, that Stable Diffusion depicts all engineers as male despite women making up around 20% of people in engineering professions. According to (Chefer et al., 2023), Stable Diffusion bears the nontrivial biases due to learned relations between not necessarily related concepts like "millennials" and "drinking".

There are recent methods focusing on diversity for Stable Diffusion generated images as well. Most of them focus on specific domains to broaden the image variability. For instance, Shipard et al. (2023) and Bansal et al. (2022) report that adding supporting context to text-to-image models prompts increases diversity in both general and human-specific fashion. In (Samuel et al., 2023) the authors report increase in generation of rare-concept images following the seed manipulation. Kim et al. (2023) trains prompt embeddings that would guide to generate images fairly according to the set of sensitive attributes, such as gender and ethnicity. The approach (Corso et al., 2023) is also unsupervised, as our approach. However, the method is not directly comparable: our work is compatible with any tool that converts a latent space with spatial coordinates to images, which includes, e.g., SDXL Turbo or other tools which do not have diffusion iterations. The strategy of (Corso et al., 2023) is entangled in the diffusion process, and thus not applicable to tools such as SDXL Turbo. Salimans & Ho (2022) suggests improving the diversity of text-to-image diffusion models by modifying the parameters of classifier-free guidance (CFG). However, this method negatively affects the quality of produced images (Fan et al., 2023).

We note that a classical method for increasing the diversity is to add text suffixes. For example, one might add texts corresponding to different skin colors or genders. However, as shown by many recent counter-examples recently (Bonyhady, 2024), this has several drawbacks, such as unrealistic outputs.

Promoting novelty and diversity is a challenge that exists both for image and text generation. In (Xu et al., 2018), the authors emphasize the importance of producing novel and diverse textual outputs. By leveraging a language-model-based discriminator, their model DP-GAN assigns high rewards to text that exhibits both novelty and fluency. Our approach for image generation also goes beyond addressing sensitive fairness concerns and opens new avenues for creative expression, as we focus not only on representation of people with different ethnicities and genders, but also, as our approach is unsupervised, on diversity of color and other dissimilarities of images in a batch.

It is common to generate multiple image versions using latent diffusion models. Even in non-sensitive scenarios, having diverse outputs is essential. A collection of similar images holds little advantage over a single image. Therefore, we propose a method that encompasses various domains, including faces, cars, animals and more to enhance diversity in text-to-image generation. By increasing diversity within batches, we can reduce the number of required generation iterations. For instance, by increasing the probability of obtaining satisfactory images from 1% with a vanilla Stable Diffusion to 5% using our method, the expected number of generated batches before satisfaction decreases by a factor 5.

---

**Algorithm 1** Generating diverse vectors in the "cap" setting

---

**Require:** batch size $B$, minimum distance $d_{\min}$
**Ensure:** Set of diverse vectors $V$

1: $V \leftarrow \emptyset$            ▷ Initialize an empty set of vectors
2: **while** $|V| < B$ **do**
3:      $v_{\text{new}} \leftarrow \text{GenerateNewVector}()$      ▷ Generate a new vector in the latent space of Stable Diffusion
4:      **if** $\text{MinDistance}(v_{\text{new}}, V) \geq d_{\min}$ **then**
5:          $V \leftarrow V \cup \{v_{\text{new}}\}$          ▷ Add the new vector to the set
6:      **end if**
7: **end while**
8: **return** $V$            ▷ Return the set of diverse vectors

---

Overall, in this article, we present Diverse Diffusion, *i.e.*, modifications to the Stable Diffusion algorithm that facilitate the generation of diverse images and thereby help to create more inclusive art within a limited number of Stable Diffusion generations (and therefore less computational power) in unsupervised fashion. Images generated with and without our approach are illustrated in Figure 1. [1]

We highlight the following contributions:

- A general unsupervised technique that can be applied to existing text-to-image models to increase image diversity, which is essential for generating realistic and varied images.
- Experiments that demonstrate the diversity advantages of our proposed approach, including against prompt-manipulation methods such as ENTIGEN (Bansal et al., 2022).

This work aims to enhance the diversity of images generated by text-to-image models, which can lead to more inclusive and representative outputs, reducing biases and promoting broader creative expression. However, our method is best suited for general concept prompts, as the quality of generated results is similar to those of the original Stable Diffusion (according to human ratings), and might not be sufficient for difficult concepts such as medical image generation.

## 2 Diversity algorithms

There are different approaches to generating diverse point sets in an unsupervised manner. For example, Latin Hypercube Sampling (McKay et al., 1979), low discrepancy methods (Niederreiter, 1992; Hammersley, 1960; Atanassov, 2004) and low dispersion. Here, we focus on low dispersion, optimized by a very limited random search for staying in the domain of validity of our latent variables. We aim to find vectors ($z_T$s) in the latent space of Stable Diffusion that are distant from one another. To accomplish this, we generate multiple vectors in the latent space until we obtain a set that contains the required number of vectors (determined by the batch size) and satisfies a specific distance requirement.

In the "**baseline**" setting, we generate images using the standard, unmodified version of Stable Diffusion without imposing any distance requirements on the latent space.

In the "**cap**" setting, we enforce a minimum requirement of $d_{\min}$ on the vectors corresponding to all pairs of images within a batch. We illustrate the procedure of choosing a set of latent vectors $V$ for batch size $B$ and minimum distance requirement $d_{min}$ in algorithm 1.

In the **"max"** setting, we impose a maximum number of iterations on searching for a new vector that would have a maximal minimal distance to all the already selected vectors in the batch. We illustrate the procedure of choosing a set of latent vectors $V$ for batch size $B$ and a maximum number of iterations requirement $N_{\max}$ in algorithm 2.

---

[1] All experiments were conducted on Jean Zay servers. Meta affiliated authors provided code and expertise and acted in an advisory role. We refrain from experimenting with pre-existing third-party generative models, such as Stable Diffusion or LDMs, and instead use a large diffusion model (2.2B parameters) trained on an internal dataset of image-text pairs.

---

**Algorithm 2** Generating diverse vectors in the "max" setting

---

**Require:** batch size $B$, maximum iterations number $N_{\max}$
**Ensure:** Set of diverse vectors $V$
 1: $V \leftarrow \emptyset$              ▷ Initialize an empty set of vectors
 2: **for** $b = 1 \leftarrow B$ **do**
 3:   $v_{\text{farthest}} \leftarrow \text{GenerateNewVector}()$
 4:   **for** $N = 1 \leftarrow N_{\max} - 1$ **do**
 5:    $v_{\text{new}} \leftarrow \text{GenerateNewVector}()$   ▷ Generate a new vector in the latent space of Stable Diffusion
 6:    **if** $\text{MinDistance}(v_{\text{new}}, V) > \text{MinDistance}(v_{\text{farthest}}, V)$ **then**
 7:     $v_{\text{farthest}} \leftarrow v_{\text{new}}$
 8:    **end if**     ▷ Find the vector with the maximum minimum distance to the existing set
 9:    $V \leftarrow V \cup \{v_{\text{farthest}}\}$           ▷ Add the farthest vector
10:   **end for**
11: **end for**
12: **return** $V$            ▷ Return the set of diverse vectors

---

In setting **"pooling_cap"** and **"pooling_max"** we apply the same exact methods as in "cap" and "max" but the distance is calculated differently. Specifically, instead of directly using the Euclidean distance, the distance is computed between the vectors after applying average pooling $8 \times 8$ to them, which down-samples the vector size to $4 \times 8 \times 8$.

We use two different settings for generating diverse image batches: **standard experiment** and **long experiment**. The long experiment is more expensive computationally, while the standard experiment has a smaller computational overhead for 'max' method. In case of 'cap' setting, everything depends on the batch size, but standard experiments are manageable for all the prompt sizes up to 50.

The parameters of a standard experiment are the following: in setting "cap" the minimal distance between latent vectors should be at least 182, while in the setting "pooling_cap" it should be at least 3.1. For algorithms "max" and "pooling_max" the number of iterations after which the farthest vector is found is 100. The choice of minimal distance, namely 182, between the latent vectors, is inspired by the average distance between 2 random vectors in the latent space. The number of iteration was impacted by the computational capabilities of the machine that we use.

The parameters of a long experiment are the following: in setting "cap" the minimal distance between latent vectors should be at least 183, while in setting "pooling_cap" it should be at least 3.1. In the setting "max" and "pooling_max", the number of iterations after which the farthest vector is found, is 10000.

The choice of settings provides variable diversity levels and variable computational complexity. In the current paper for small batch sizes $(3, 5, 10)$, we create both standard and long experiments, and for the big batch sizes $(50)$ we create only standard experiments. In the cap and pooling cap setting, due to the distance limitations and no limit on a number of iterations, the experiment for big batches becomes increasingly slower. That is why, for purposes of limiting computational cost, we recommend the "pooling_max" method for batch sizes more than 50. We note that, for the 'max' method in standard setting, we spend less than 3s/image. This is negligible compared to the image generation cost, e.g. on a MacBook Pro M1 for all cases considered in the present paper.

# 3 Evaluation Methods

In order to evaluate the diversity of generated images, we use both quantitative and human evaluation methods. For quantitative methods, we focus primarily on color diversity image similarity metric LPIPS. Increasing image color diversity is an important aspect of ensuring general diversity of the images: we would like to get representation for people, animals, and objects of all colors.

### 3.1 Color diversity

To assess the color diversity in an image batch $b$, we employ a method that involves extracting color information from each image in the batch using the RGB color model, which represents colors as a combination of red, green, and blue channels. Specifically, we compute the mean value for each channel $(R_i, B_i, G_i)$ in the given image $i$, and identify whether one of these colors is predominantly present in the image $i$.

To determine the dominant color in image $i$ with respect to the coefficient $K$, we define the image as having a dominant color of Red if $R_i > K \times max(G_i, B_i)$, as Green if $G_i > K \times max(R_i, B_i)$, as Blue if $B_i > K \times max(G_i, R_i)$, and as None if none of these inequalities are true. We denote the dominant color of image $i$ with respect to the coefficient $K$ as $D_K(i)$.

To evaluate color diversity of an image batch $b$, we compute the number of dominant colors $N_K(b)$ in that batch with respect to the coefficient $K$.

$$N_K(b) = 3 - \prod_{i \in b}(1 - I(D_K(i) = Red)) - \prod_{i \in b}(1 - I(D_K(i) = Green)) - \prod_{i \in b}(1 - I(D_K(i) = Blue)), \quad (1)$$

where $I$ is an indicator function.

The first color diversity metric, across a set $B$ of batches, is an average number of dominant colors present in the batches of that set with respect to the coefficient $K$.

$$Avg_K(B) = \frac{\sum_{b \in B} N_K(b)}{||B||}, \quad (2)$$

where $||B||$ is the total number of batches in the set $B$.

The second color diversity metric aims to compute the proportion of batches that contain at least one image predominantly exhibiting red, blue or green color (we want to have each of the three colors to be dominant at least once in the batch), considering RGB image encoding. This allows for the quantification of color variability within a batch. Specifically, for various values of the coefficient $K$ and a set $B$ of batches, we compute the proportion

$$C3_K(B) = \frac{\sum_{b \in B} I(N_K(b) = 3)}{||B||}. \quad (3)$$

Another color diversity metric is the proportion of batches containing images with at least 2 different dominant colors (*i.e.*, not all the images in a batch are predominantly red, blue or green). Specifically, for various values of the coefficient $K$ and a set $B$ of batches, we compute the proportion

$$C2_K(B) = \frac{\sum_{b \in B} I(N_K(b) \geq 2)}{||B||}. \quad (4)$$

For most criteria and in particular for proportions of batches, the greater the proportion of batches the better.

### 3.2 LPIPS metric

Similarly to (Ham et al., 2023) and (Huang et al., 2022), we use LPIPS (Learned Perceptual Image Patch Similarity) (Zhang et al., 2018) to evaluate the diversity of the images generated by Stable Diffusion. We measure the pairwise similarity between images in a batch and compare the obtained values for different modifications of the Stable Diffusion generation algorithm. Note that LPIPS is sometimes used as 1-LPIPS (a.k.a LPIPS loss), making discussions confusing: we use the LPIPS form in which lower values correspond to greater diversity.

### 3.3 Ethnicity and gender classification for images portraying humans

As mentioned in (Theron, 2023), Stable Diffusion may lack diversity in ethnicity representation. That is why, for the prompts that we use for the human face generation, we compare ethnic diversity in between

our methods and basic version of Stable Diffusion. In order to identify the ethnicity of a person present on an image, we use DeepFace ethnicity recognition (Taigman et al., 2014). In particular, we identify the following groups of ethnicities: (i) Black, (ii) Asian, (iii) Hispanic, (iv) White or Middle Eastern. For gender classification (male/female), we also use DeepFace.We compute the percentage of batches where all pairs of genders and ethnicities are present or at least 3 out of 4 ethnicities are present (similarly to colors).

### 3.4 Multiplicative improvement

Our method aims to increase representation of underrepresented groups. That is why for all the metrics mentioned above, we compute percentage versus multiplicative improvement score. Here percentage stands for the percentage of batches that follow some characteristic $C$ (for example, contain images of Asian men) and multiplicative improvement stands for multiplicative increase in this percentage for our preferred method (**"pooling_cap"**) compared to the baseline method (**standard Stable Diffusion**). By using this metric, we can evaluate our efforts in promoting the inclusivity of underrepresented classes.

## 4 Experiment setting

As a baseline, we use Stable Diffusion 1.5 with PNDM Scheduler that generates $512 \times 512$ images. We measure diversity using LPIPS or artificial classes (image hue) or human-centered criteria (gender/ethnicity) and check various batch sizes. We use machines with 8 Tesla V100-SXM2-32GB GPUs and 80 x86_64 CPUs. The full code is (anonymously) provided in https://anonymous.4open.science/r/DiverseDiffusion-1012. See supplementary material for reproducibility details.

### 4.1 Experiments on small batches

For our small batch experiment setup, we choose the following list of prompts:

- "face",
- 'rose",
- "butterfly",
- "cat",
- "horse",
- "car",
- "ornament",
- "bird",
- "color",
- "a professional photograph of an adult person face",
- "photo of an animal in the grass" and
- "octane, hyperrealistic, backlit".

In each experiment, we consider batches of 3, 5 and 10 images in order to compute diversity metrics in each of these batches, and to compare our modifications with original Stable Diffusion. For each batch size and each modification, we create at least 2500 batches.

### 4.2 Experiments on big batches

For our big batch experiment setup, we choose the following list of prompts:

- "a professional photograph of a man face",
- "a photograph of a person with different colored eyes",
- "a passport-style photograph of a person's face",
- "a professional photograph of an adult person face",
- "a close-up photograph of an elderly person's face" and
- "a beauty shot of a model's face".

These prompts are all centered on human photos and thus allow us to evaluate not only LPIPS and color diversity but also gender and ethnicity variation, which are crucial to ensure in any human-centered applications of diffusion models. In each experiment, we consider batches of 50 images in order to compute diversity

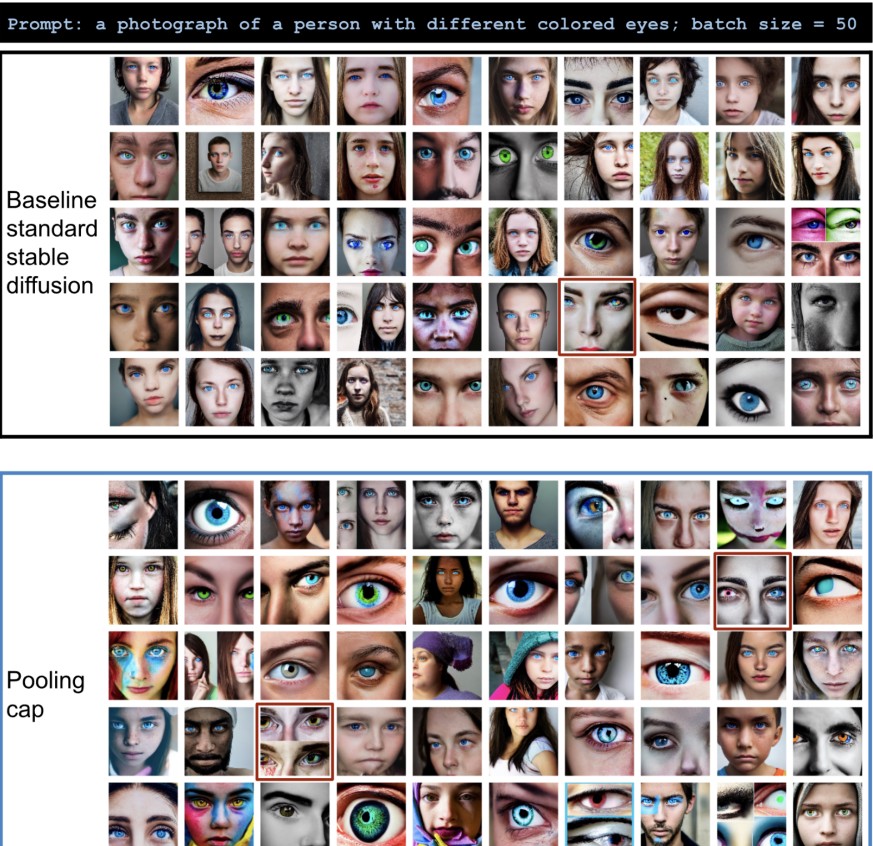

Figure 2: Examples of images generated with the prompt "a photograph of a person with different colored eyes": pooling_cap against baseline Stable Diffusion, batch size=50. Images corresponding to the "expected output" of the prompt are highlighted in red.

metrics in each of these batches and compare our modifications with the baseline: original Stable Diffusion. For each modification, we create at least 900 batches.

# 5 Experiments and results

An example of image batches generated with our method "pooling_cap" and original Stable Diffusion is presented in Figure 2. In two randomly chosen examples among all batches generated by the baseline and "pooling_cap" for the prompt "a photograph of a person with different colored eyes" and batch size 50, we can see that while getting images corresponding to the prompt remains difficult for our chosen method, we still get improvement both in number of images corresponding to the prompt and ethnic diversity.

Further, we present the experimental evaluation of our proposed methods for promoting diversity in generated image batches. We assess the color diversity, gender and ethnicity representation, and image diversity using the LPIPS metric. The experiments are conducted on various prompts and compared against the baseline Stable Diffusion method.

## 5.1 Color evaluation

In this subsection, we evaluate the color diversity of the generated image batches. We present multiplicative improvement results, summarizing the color diversity improvement for various values of the coefficient $K$ across different prompts specified in Section "Experiments on small batches". Other experiment results are provided in the supplementary material.

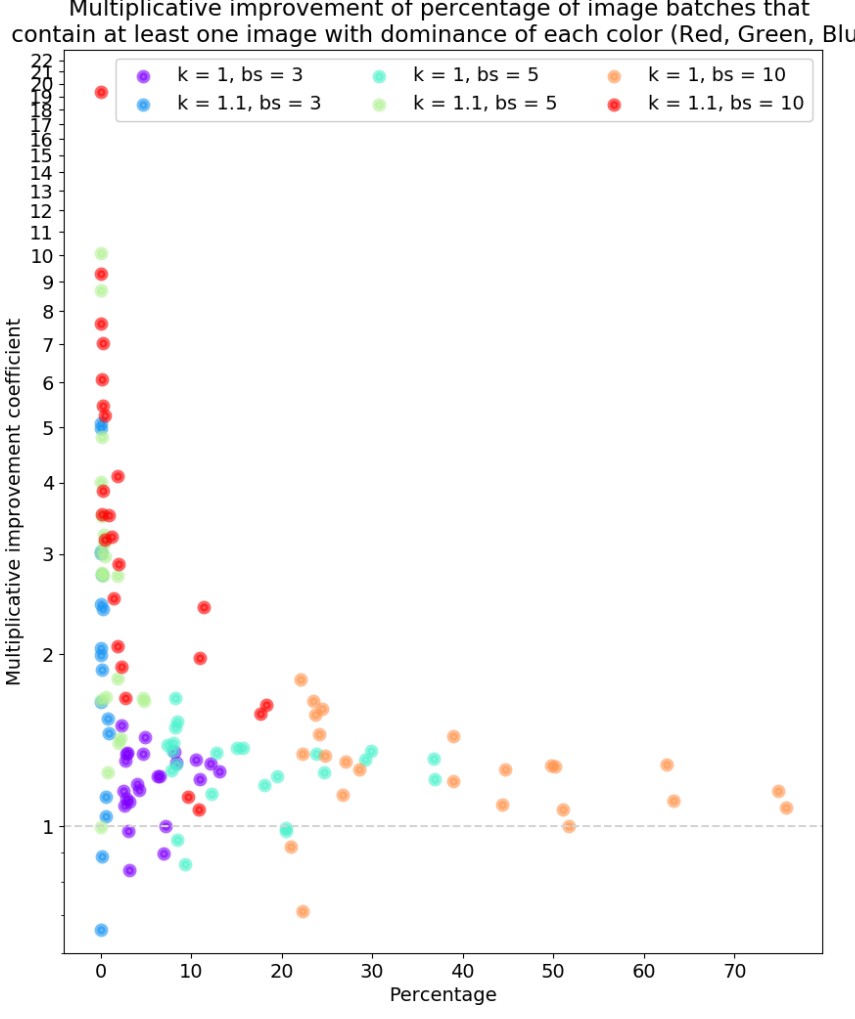

Figure 3: Multiplicative improvement of percentage of batches containing images with all 3 dominant colors: pooling_cap against baseline Stable Diffusion, depending on $K$ as specified in the text and batch size $bs$. Improvements are greater on the left, *i.e.*, more difficult cases.

In Figure 3, we can see that in majority of the cases using pooling_cap method does not decrease the representation of batches with all 3 colors dominance. We also can see that we get very high ($> 2.5$) improvement coefficient numbers in cases where color dominance is defined using coefficient k = 1.1, which significantly increases the number of batches featuring this rare characteristic.

In Figure 4, we can see that in majority of the cases using pooling_cap method does not decrease the representation of batches featuring at least 2 dominant colors. Similar to figure 3, we observe very high ($> 2.5$) improvement coefficient numbers only in cases where color dominance is defined using coefficient $K > 1$, which significantly increases the number of batches featuring this rare characteristic. We also can see that for modes where having at least 2 dominant colors per batch is a rare characteristic (less than 50% of batches feature it), we always have improvement coefficient $> 1$, thus increasing the color diversity.

## 5.2 Gender and ethnicity evaluation

Here, we evaluate the impact of our proposed method on the diversity of gender and ethnicity representation in the generated image batches.

Figure 4: Multiplicative improvement for the percentage of batches containing images of at least 2 of the 3 categories: the improvement is greater for more difficult cases, which are on the left (pooling_cap against baseline Stable Diffusion method).

Our results, as illustrated in Figure 5, demonstrate a remarkable enhancement in the representation of underrepresented categories. This improvement is consistently observed across all ethnicity/gender pairs that initially appeared in less than 60% of the batches. Notably, our cap pooling technique has led to a substantial increase in their presence, with certain categories being up to 2.4 times more prevalent than in the standard Stable Diffusion version.

While we see some detrimental results for Hispanics representation, it is important to note that ethnicity identification based solely on a person's image is not always accurate for artificial intelligence methods. In this research, we primarily focus on the diversity between white and black individuals, as they can be visually distinguished with minimal ambiguity (Maluleke et al., 2022). In Figure 5, we observe a positive improvement in the representation of black individuals, both male and female, compared to the baseline results for Stable Diffusion.

Overall, these findings highlight the effectiveness of our approach in promoting diversity and addressing underrepresentation in gender and ethnicity across various image batches.

### 5.3 LPIPS evaluation

In this subsection, we evaluate the diversity of generated images using the LPIPS (Zhang et al., 2018) metric. LPIPS measures the perceptual similarity between images based on deep neural network representations. A lower pairwise LPIPS loss score (where the LPIPS loss is traditionally $LPIPS_{loss} = 1 - LPIPS_{score}$) indicates greater diversity among the generated images.

Here we compare the performance of different methods, including the baseline and our proposed "pooling_cap" method, across different batch sizes.

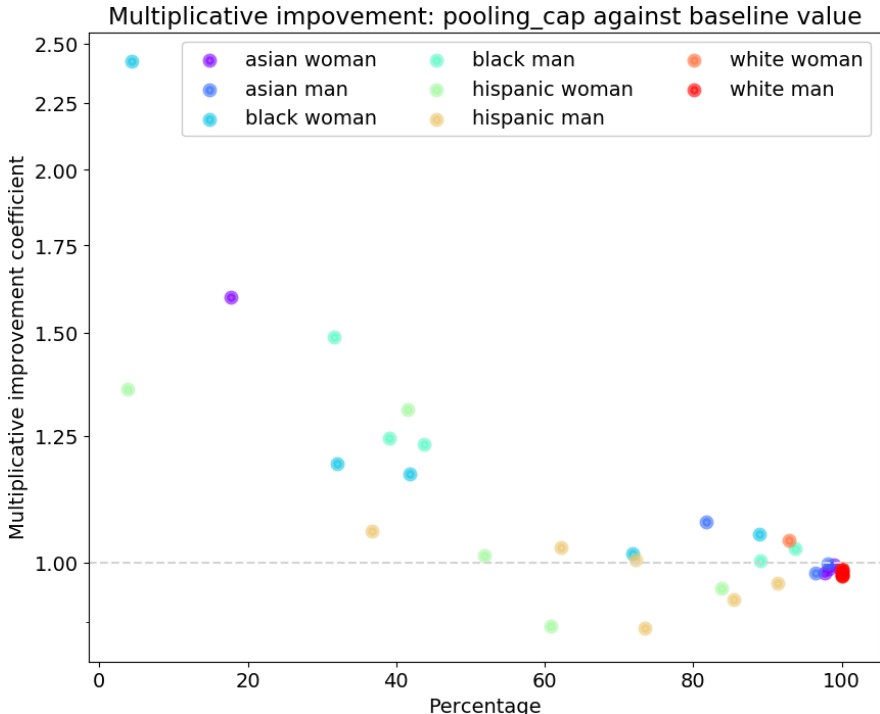

Figure 5: Multiplicative improvement (increase) of percentage of batches containing various ethnicity + gender pairs: we get a greater improvement for more difficult cases (pooling_cap against baseline Stable Diffusion method).

Figure 6 presents the average batch pairwise LPIPS distance for a batch size of 3. We can see that, for most of the experiments (though not all), the "pooling_cap" method proves to be more diverse than the baseline, while for certain cases such as "face", the baseline outperforms "pooling_cap". We also notice that "pooling_cap" is not the only method that performs better than the baseline. Others, such as "cap", prove to be the best for various prompts as well. For example: "rose", "ornament" and "car". We now switch to greater batch sizes, for which results are clearer.

Figure 7 illustrates the average batch pairwise LPIPS distance for a batch size of 5. We can see that, for most of the experiments, the "pooling_max" method proves to be the best for diversity enhancement. "Pooling_cap" is also better than the baseline for most of the cases. Similarly to Figure 6, we can also notice that for the cases where baseline had a small average LPIPS loss distance (for instance "octane" and "photo of an animal"), all the proposed methods significantly outperform the baseline. This shows that even for small batches, our methods provide diversity benefits in cases where it was really lacking.

Figure 8 showcases the average batch pairwise LPIPS distance for a batch size of 10. Here, we can see that, for most of the experiments, "pooling_max" is the most diverse among all the methods, and "pooling_cap" is also better than the baseline for most of the cases. We again notice that in cases where the baseline had a small average LPIPS distance (for instance "octane"), all the proposed methods significantly outperform the baseline. This shows that even for batches of size 3 to 10, our methods provide diversity benefits in cases when it was really lacking. In contrast to the results presented in Figure 6, in Figure 8 "pooling_max" method shows to always outperform the baseline. This observation indicates that our methods provide better diversity guaranties for the bigger batch sizes.

In contrast to the results shown in Figures 6, 7 and 8, in Figure 9 we can see that the "pooling_cap" method is consistently highlighted as the most diverse across all experiments. This emphasizes that larger batch sizes lead to a clearer benefit of our method: not only for hard cases with very poor LPIPS, but for all

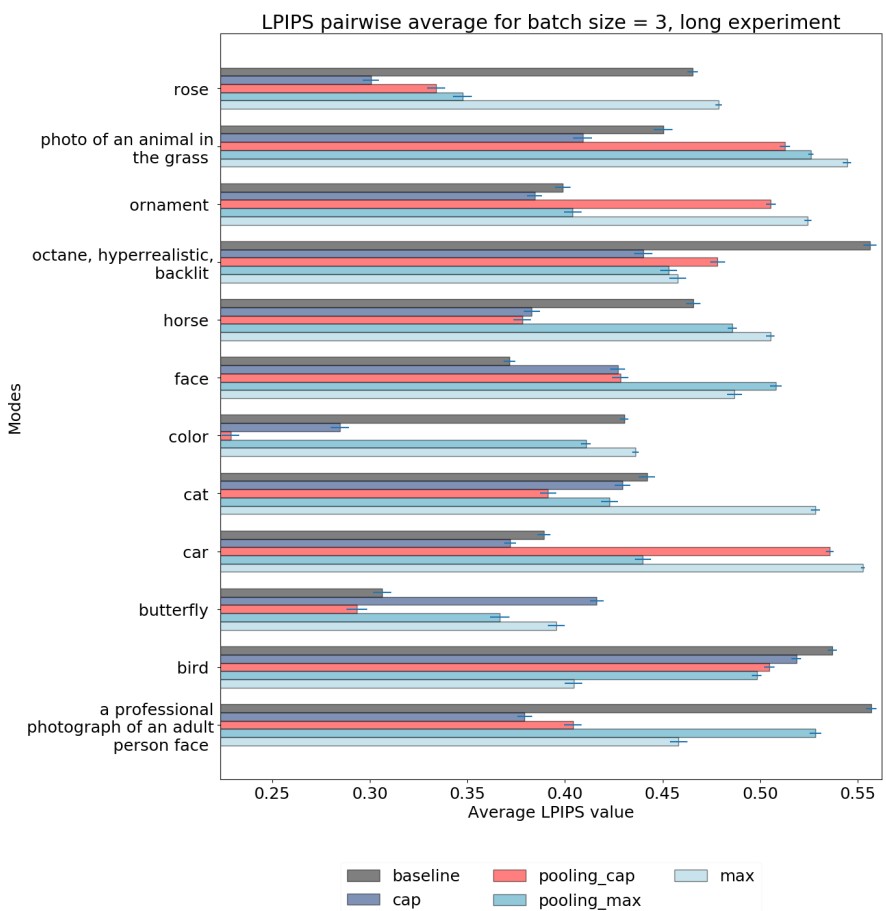

Figure 6: Average batch pairwise LPIPS loss (the lower the better), batch size=3: no clear conclusion overall, due to the small batch size.

| Method | average LPIPS loss |
|---|---|
| Baseline | $0.354 \pm 0.004$ |
| ENTIGEN | $0.325 \pm 0.004$ |
| **Pooling_cap** | **$0.294 \pm 0.005$** |

Table 1: Average pairwise LPIPS loss value across different batches with the same prompt. Baseline = Stable Diffusion with random choice of latent vector, ENTIGEN = prompt-manipulation diversity method, Pooling_cap = our proposed method.

cases. This experiment strengthens our earlier observations on color diversity and solidifies the superiority of the "pooling_cap" variant over the standard Stable Diffusion approach in terms of diversity.

For the experiment presented in Table 1, we compute the average LPIPS distance between pairs of generated images. The batch size is 50, we compute this average over the union of all generated batches. The prompts are those in Section 4.2. We do this experiment across standard Stable Diffusion, our preferred method "pooling_cap", and a prompt-manipulation diversity method known as ENTIGEN (Bansal et al., 2022). More precisely, to each prompt, ENTIGEN adds either "irrespective of their gender" or "irrespective of their color". The ENTIGEN approach is specifically designed to eliminate human-centered biases such as gender and ethnicity. We can see that even in this real-world context, our agnostic method (which ignores gender and ethnicity, and focuses on numerical metrics) achieves a better diversity than both vanilla Stable Diffusion and ENTIGEN.

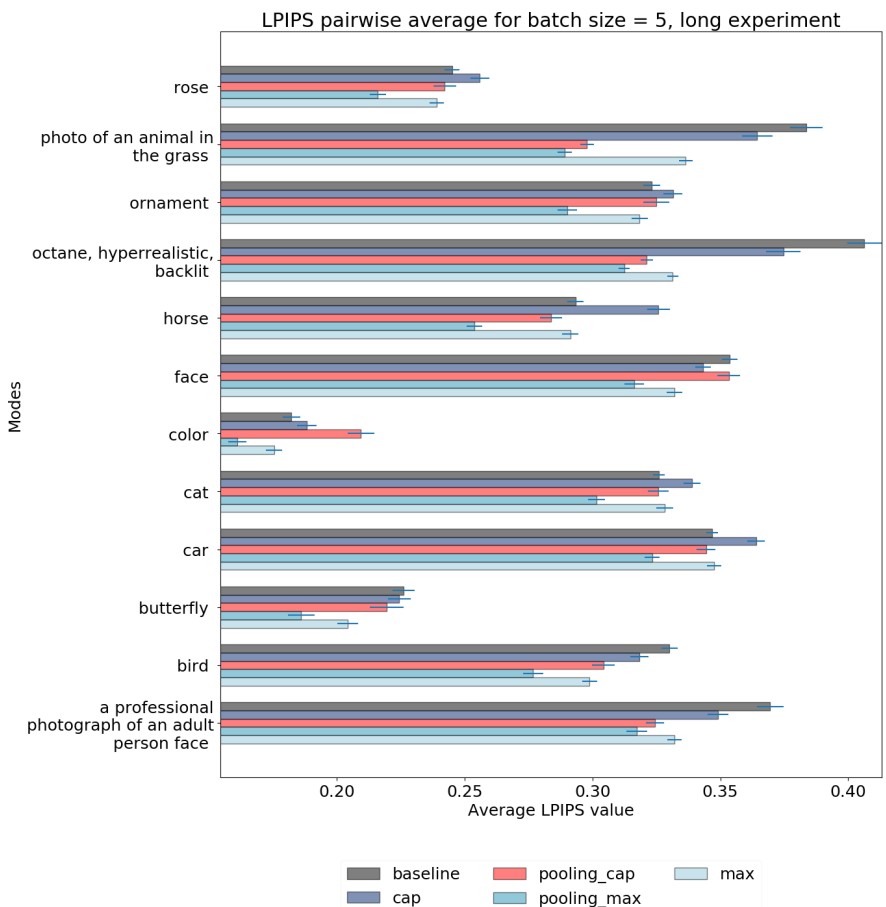

Figure 7: Average batch pairwise LPIPS distance (the lower the better), batch size=5: the two Pooling methods are often better than the baseline, though results are clearer for batch size 50.

| Prefer vanilla SD | No preference | Prefer *Pooling_cap* |
|:---:|:---:|:---:|
| 35.8% | 25.6% | 38.6% |

Table 2: Results of the human study in terms of quality of individual images. Each question corresponds to a prompt randomly drawn in $\{bird, rose, butterfly\}$. We present one image generated by vanilla SD, and one image generated by $Pooling_{cap}$. The batch size is 3 for butterflies, 5 for roses and 10 for birds. The images are randomly positioned left/right, and the question is "For prompt XXX, do you prefer the image on the left, on the right, or no preference". We observe a result ($51.4\% \pm 1.5\%$, counting 1 when $Pooling\_cap$ wins, 0 when it loses, and 0.5 for draw) slightly above 50% (not statistically significant) in favor of our method, suggesting that there is no loss of quality. Six distinct human raters, not professional, not working in the computer vision field, double-blind.

## 5.4   Quality assessment

In Table 2, we report the results of a human study. Unlike classifier-free guidance (CFG) (Salimans & Ho, 2022) that is currently used for text-to-image diffusion models diversity improvement, our proposed method Pooling_cap does not decrease the quality of generated images. While we provide the same image quality (up to an error), we significantly improve the diversity of generated images: as we show in the previous section, we outperform prompt-manipulation methods such as ENTIGEN. Thus, our method Pooling_cap doesn't have quality/diversity trade-off, and can be applied to different text-to-image models, for diversity enhancement.

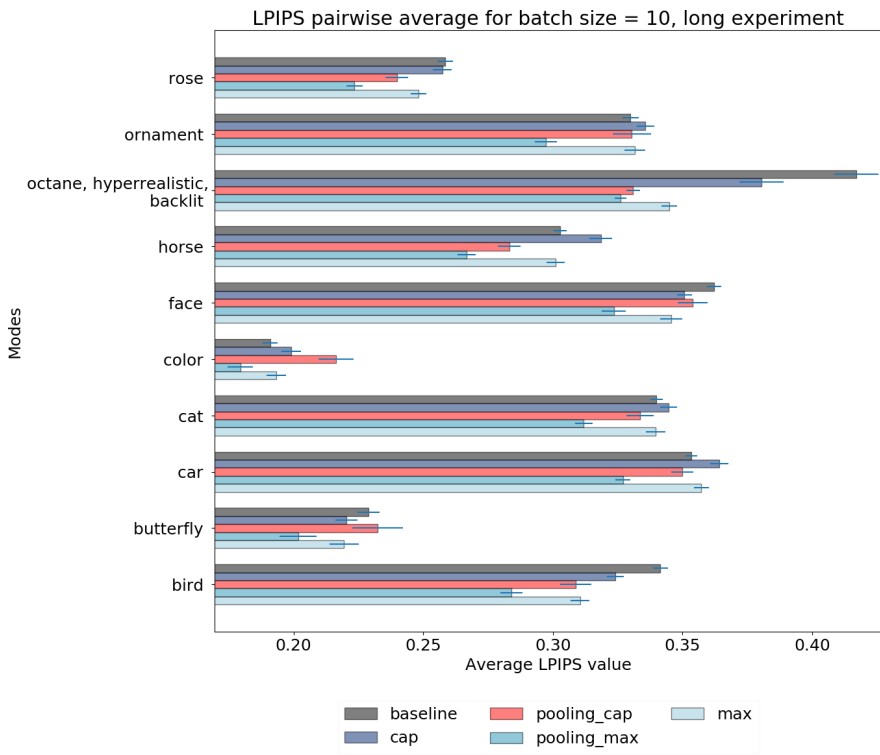

Figure 8: Average batch pairwise LPIPS loss (the lower the better), batch size=10. The two Pooling methods are often better than the baseline, though results are clearer for batch size 50.

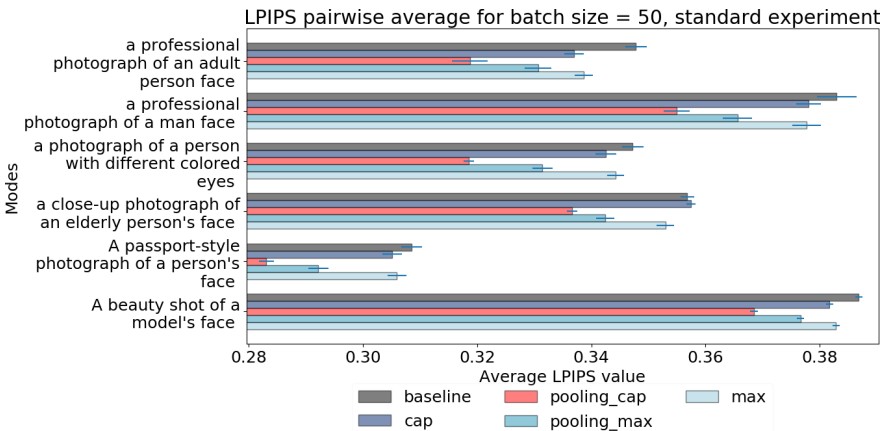

Figure 9: Average batch pairwise LPIPS loss (the lower the better), batch size=50. For this batch size, in all cases pooling methods (both max and cap) are beneficial to diversity as measured by LPIPS.

## 6 Conclusions

In conclusion, our contributions include

- a general technique that can be applied to existing text-to-image models to increase image diversity, which works in an unsupervised manner with a negligible overhead and
- experiments that showcase the diversity advantages of our proposed approach applied to Stable Diffusion, through classification (both image hue and gender/ethnicity) and LPIPS measurements.
- experiments proving that our approach maintains the quality of images (human raters).

Experimental results highlight the impact of our proposed method on color diversity. By analyzing the multiplicative improvement in the batches containing images with dominant colors, we observe that the "pooling_cap" method consistently maintains or increases the representation of batches with all three dominant colors. Additionally, we notice significant enhancement when using the color dominance coefficient $K > 1$ (*i.e.*, cases in which success is rarer, hence the problem is more difficult). These findings validate the effectiveness of our approach in promoting diverse color compositions in generated images.

Our experimental results consistently demonstrate the superiority of our "pooling_max" and "pooling_cap" methods over the baseline (standard Stable Diffusion) in terms of average pairwise LPIPS distance within batches. This evaluation metric serves as a reliable measure of image diversity, and our methods consistently outperform the baseline, showcasing their ability to generate more diverse images. These findings highlight the effectiveness of our approach in expanding the range of image variations, and improving overall diversity.

Furthermore, the results show a notable improvement in the representation of underrepresented categories across different ethnicity/gender pairs. The "pooling_cap" method led to a substantial increase in the presence of these categories, with some categories being up to 2.4 times more frequent, compared to the baseline. It is important to acknowledge the inherent limitations of ethnicity identification based solely on visual cues, but our focus on improving the representation of underrepresented categories aligns with the goal of promoting diversity and inclusion in image generation.

Additionally, we show that we do not decrease image quality while providing superior diversity (compared to prompt-manipulation methods such as ENTIGEN and vanilla Stable Diffusion), as shown by the human study in Table 2 and LPIPS evaluation in Table 1.

In the future, we intend to assess the generalizability of our approach, by applying it to different text-to-image models. This will allow us to determine the extent to which our technique can be adapted and utilized in various contexts. One of our methods is already available as a plugin for a main image generator with more than 100000 github stars[2]. Our ultimate goal is to advance the field of text-to-image generation by pushing the boundaries of diversity and realism, enabling the creation of visually varied and inclusive images for a wide range of applications.

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
