# OpenReview forum: "Diverse Diffusion: Enhancing Image Diversity in Text-to- Image Generation"
_TMLR — Rejected by TMLR_

### Review · Reviewer_BA7d · 2024-07-18

**Summary Of Contributions:**

The authors propose a method for generating more diverse samples from pretrained latent diffusion models, by resampling noise vectors until these have some minimum distance to the set of existing noise vectors. They propose variations for selecting maximum distance vectors, e.g. settings max, cap simply compute distance in latent space, and pooling_max and pooling_cap average pool first.

The authors show that their approach leads to increased color diversity, improved diversity according to batch-wise LPIPS distance, and diversity in gender and ethnicity for face generation.

**Audience:**

Yes

**Broader Impact Concerns:**

No concerns. This method can produce more diverse images, which is potentially positive. There are no guarantees that it will, but the authors do not make any strong claims here.

**Claims And Evidence:**

No

**Requested Changes:**

The technique is simple, if it's an easy plug-in method to generate more diverse outputs, that will be interesting for the TMLR audience.  I believe a (much) stronger empirical study is needed to provide evidence that this technique delivers on its promise, for example using a larger set of prompts, ideally from a benchmark captioning dataset. This should be feasible with 8 V100s.

The scores should be aggregated, for example in tables, so that the conclusion becomes easy to understand. Comparing scatter plots is hard, especially when these points do not share the same values on the x-axis.

Please add experimental details:
- more information on the evaluation procedures, meaning of percentage of batches by giving intuitive example would help
- number of random seeds for each setting and error metric should be mentioned when errorbars are provided. I assumed this was equal to the number of batches, and that this is mean + standard deviation.
- image resolution (I'm assuming 512, given that 8x8 pooled latents are 8x8)
- section 4 mentions StableDiffusion v-5 -- should probably be 1.5?

Better formatting would also make the paper easier to read. The generated images are the main selling point, so they can be much larger. You can also use subfigures to make the paper more compact.

**Strengths And Weaknesses:**

I do think guaranteeing diverse images while retaining high quality is challenging, as some prompts are so specific that a standard stablediffusion model will output very similar images. It is commendable that this work aims to address that.

I believe method can be summarized as "resample latent vectors until some distance in latent space is met". This is simple, and the simplicity of a method can be an advantage. However, this method comes without guarantees about the effects in image-space. A distance in noise vector space may have a very different meaning for one prompt than for another prompt.

I do not think the shown evaluation procedures are convincing as is.
- The evaluation in this work is fairly limited:
    - only 11 prompts are used for small-batch experiments, and 6 for big batch experiments. The evaluation would be much stronger if an image captioning dataset was used.
    - except for Table1, this work only compares to base stablediffusion. A comparison against one of the related works, especially one of the stronger baselines, could give us an additional reference point to tell how well this method works in practice. If DiverseDiffusion is competitive with a much more involved method, that makes it a strong contender. This is hard to judge currently. There is only one comparison to ENTIGEN in Table 1, and ENTIGEN is quite a simple method as well, only adding to the prompt if I understand correctly.
- The evaluation metrics are not well-described, which makes the scatter plots hard to interpret.
    - How do you compute the percentage of batches number exactly? What does it mean if this percentage is 50% for example?
    - Multiplicative improvement hides information: the absolute values. A multiplicative improvement from 2.0 to 1.0 is potentially more impressive than an improvement from 0.002 to 0.001, depending on the metric. Is it not possible to report the absolute pairwise LPIPS and color numbers in a table for a fixed number of batches and prompts?
- A user study is conducted, which finds no noticeable difference between base SD and the pooling_cap method. However, this only measures the image quality, and not the diversity of generations, which is the selling point of the method. For quality assessment, a metric such as FID or even prompt-based CLIP score could provide additional evidence that the image quality does not degrade.

Additionally:
- The pooling-based variants have a specific consequence: averaging 8x8 gaussian noise vectors will in expectation result in a zero output. This smoothens out any differences between sampled noise vectors. I would expect that maximizing distance in the pooled output will therefore emphasize dissimilarity of blocks of 8x8. I believe some intuition about why this is desirable or advantageous could help.
- Errorbars are reported for some barcharts, how many random seeds are these based on?

In my opinion, Figure 6 provides the strongest evidence that the method works: it results in increased diversity as measure by LPIPS, consistently across prompts and for a large batchsize. Perhaps a point in between batch size 5 and batch size 50 can shed light on when this method works best?

Without theoretical guarantees or a strong empirical evaluation, I find it hard to argue in favor of acceptance.

---

### Review · Reviewer_o2su · 2024-07-20

**Summary Of Contributions:**

The paper proposes Diverse Diffusion, a method to generate diverse images by sampling a set of diverse vectors from the latent space.  Two main strategies are proposed for constructing this set by iteratively sampling from the latent space:
  1. "cap" setting: only adding a new vector to the set of selected vectors if the distance of the new vector to the set is above a minimum distance threshold
  2. "max" setting: sample up to maximum number of iterations, and then add the vector that is the maximal distance from the existing set
For each strategy, either the Euclidean distance is used, or a vectors are pooled and then the distance is taken.
Experiments are conducted with small batches (less than 10 images generated per prompt, over 12 prompts) and large batches (50 images generated per prompt, over 6 prompts), with diversity is measured using number of colors, evaluation of gender and ethnicity, and pairwise LPIPS.  Results on the show that the proposed method provides more diversity than the baseline and one prior work (ENTIGEN).

The main contribution of the work are:
1. Proposed method to sample diverse vector to obtain diverse images
2. Experiments comparing different settings

**Audience:**

No

**Broader Impact Concerns:**

The work should add a Broader Impact Statement both on the positive (e.g. generated image could be more diverse) and negative impact (e.g. generated images may not match the input prompt more than the baseline) of this work.

**Claims And Evidence:**

No

**Requested Changes:**

Significant revisions and experiments are required before the work would be of interest to the community
1. Additional baselines should be considered.
  - For instance, the paper states that "Salimans & Ho (2022) suggests improving the diversity of text-to-image diffusion modelsby modifying the parameters of classifier-free guidance".  This suggests a natural comparison point.  Even if it "negatively affects the quality of produced images", it should emprically be compared against.
  - Similarly, other methods mentioned such as seed manipulation (Samuel et al. 2023), methods limited to diffusion models (Corso et al. 2023) should be compared against as the experiments are with a diffusion based model (Stable Diffusion).
2. There should also be more evaluation of text-to-image alignment as well as generated image quality in addition to diversity.
3. Experiments should be expanded to be more systematic and on a much larger set of prompts.  A potential starting point would be evaluate on existing benchmarks of prompts.  For instance, ENTIGEN provides 246 different prompts (36 neutral).
The following benchmark can also be useful for a larger set of prompts (~45K) and evaluating text-to-image fidelity / alignment.
  [1] HRS-Bench: Holistic, Reliable and Scalable Benchmark for Text-to-Image Models
4. Experimental results should be aggregated and presented in so it is easier to see the how the different variants compare
  - While various results are presented, it is hard to determine from the Figures (esp Fig 6-9), which method is more diverse
5. Discussion of tradeoff of quality and diversity needs to be improved
   It is hard to believe the claim "our method Pooling_cap doesn’t have quality/diversity trade-off".  Intuitively, there should be a tradeoff, and just qualitatively from Figure 1 and Figure 2, there is a tradeoff.  The proposed method generate images that either does not match the prompt or is lower quality.
6. Writing should be improve to clarify some details
  - The description of the pooling distances was somewhat confusing.  It was not clear to me what the pooling was applied over.
  - There is description of "standard experiment" and "long experiment" setting.  The rationale for the two are not specified and it is not clear how they impact experimental results.
  - It's no clear how the parameter settings affect performance
7. Introduction need to be improved to have a more to the point and clear about the focus of this work.

    The abstract states that "we introduce Diverse Diffusion, a method for boosting image diversity beyond gender and ethnicity, spanning into richer realms".

  - However, the introduction starts with discussion of various concerns such as the use of text-to-image models in sensitive applications, misuse of text-to-image generation.  It's unclear how these two points relate to diversity.
    - The introduction states that "there have been concerns regarding potential causes of their [text-to-image generation models] usage for various sensitive applications such as medical imaging".  It's unclear what "potential causes" refers to, but how does the use in sensitive applications (such as medical domain) relate to diversity - is it that there are issues because it is desirable to generate diverse (potentially inaccurate) images, or is are there concerns because the images that are generated are not accurate due to quality (e.g. lung xrays that are generate do not represent real world lung xrays) or image -to-text fidelity (e.g. lung xrays with tumors do not have tumors).
    - Regarding the paragraph about "the potential misuse of text-to-image generation", it was not clear how that is related to this work.
    - Regarding the paragraph on "bias can arise from various sources", while logically one can connect bias to lack of diversity, this paragraph fails to actually make the connection.  It describes "increasing training data quality" and "ensuring that diffusion-based methods do not amplify any biases present in the training data".  There is no actual text connecting those points to diversity.

  - Once the introduction shifts into diversity, the introduction makes more sense, but there is quite a bit of focus is on bias and methods that tackle culture diversity.  Thus is it not that clear whether the paper is intending to focus just on diversity due to bias or more generally (as stated in the abstract).  In addition, the only comparison to prior work on generating diverse images is with ENTIGEN, a benchmark dataset that is targeted for evaluating text-to-image generation bias with gender/culture.

8. Minor typos
  - Page 2. Incomplete sentence / floating reference: Fan et al. (2023)

**Strengths And Weaknesses:**

Strengths
- The idea of sampling diverse vectors to get more diverse image generation results is simple and intuitive


Weaknesses
- Experiments are weak.
  - The proposed method is evaluated on a very small number of prompts.
  - There is limited comparisons against alternative methods.  The only comparison was against ENTIGEN, which is designed for reducing biases of gender, ethnicity, and culture, and the augmentations to the prompts are not necessarily tailored for more general prompts.
  - There is very limited evaluation of whether text-to-image alignment is preserved and the quality of the image generated.  There was a human study for generated image preference for given prompt, and the quality of the image generated is likely not impacted by the proposed method.  However, the human study was very underwhelming.
     - The human study was run on just three prompts (bird, rose, and butterfly) with just three human participants
     - Users were shown two images and asked which one they preferred.  There was no attempt to separate the notion of the text-to-image alignment (fidelity) from the image quality.
- It is unclear whether generated images are faithful to the input text (especially when the text input is complex).
  - Even in the simple case ("rose" for Figure 1, it is unclear whether some of the images generated by "Diverse Diffusion" is desirable for the input "rose" as one is a person with two roses, and the original baseline seems to be generating a diverse collection of more high quality roses)
  - It's also not clear why having many dominant colors is necessarily always good - it may be diverse in color, but that doesn't mean it is faithful to the original text input.  For instance, if the text input was "red roses against a red backdrop", then it is not desirable to have color diversity.
- Logic of the writing (especially the introduction) is poor, with mentions to various prior work without logically connecting them to the focus of this work.

---

### Review · Reviewer_SchA · 2024-07-28

**Summary Of Contributions:**

This paper aims to use the latent diffusion model to generate images with diversity through text guidance. The main idea is to find a series of latent vectors with high distance and generate different with these vectors. Experimental results show that the proposed method can produce more diverse images than the baseline. However, the novelty is limited, the method is not introduced clearly, and the experiments are not sufficient. I think it's not ready for publication,

**Audience:**

Yes

**Claims And Evidence:**

No

**Requested Changes:**

See the weaknesses above.

**Strengths And Weaknesses:**

Strengths:
1. The paper is easy to follow.
2. Visualization in Fig. 1 and Fig. 2 shows that the proposed method generates more diverse images than the baseline.
Weaknesses:
1. To generate diverse vectors in latent space is too straightforward, which decreases the paper's novelty.
2. It lacks the comparison between the proposed approach and the SOTA ones. A lot of approaches are introduced in the introduction section, but only ENTIGEN  is compared.
3. It's not clear how to use the diverse vector V.
4. What's the best setting of the proposed approach and why?

---

### Decision · Action_Editor_fP35 · 2024-09-08

**Recommendation:** Reject

**Comment:**

The paper is not suitable for publication given the issues with the claims and evidence. Reviewers noted that this resubmission only involved minor changes to text, and did not include the fundamental and necessary changes to the evaluation setup that would be required to consider publication.

**Audience:**

One reviewer pointed out that the topic is relevant for the TMLR audience given general interest in improving diffusion models.

**Claims And Evidence:**

Reviewers generally agreed that the claims in the paper were not supported by convincing evidence. In particular, all reviewers pointed out that the evaluation was insufficient - i.e., the proposed method is compared to a limited set of methods (and, in particular, is not compared to state-of-the-art methods), on a small set of prompts, and without consideration for faithfulness. Reviewers also noted that the results were unclear and hard to understand.